## Research Article

anti-social media exposure; peer victimization; teacher victimization; school-based victimization; subjective well-being

**Corresponding author:**
Tosin Yinka Akintunde;
Email: akintundeolayina84@gmail.com

# Modeling the pathways from antisocial media exposure to subjective well-being through school-based victimization in Nigeria

Tosin Yinka Akintunde[1] [iD], Stanley Oloji Isangha[2], Derrick Ssewanyana[3], Olufunto O. Adewusi[4], Temitayo Kofoworola Olurin[5], Stephen Nkah Akongnwi[6] and Oluseye David Akintunde[7]

[1]Department of Sociology, Hohai University, Nanjing, China; [2]Social and Behavioral Sciences, City University of Hong Kong, Hong Kong; [3]Utrecht University, Netherlands; [4]University of Alberta, Canada; [5]Federal University of Agriculture Abeokuta, Nigeria; [6]University of Buea, Cameroon and [7]Changchun University of Science and Technology, China

## Abstract

The pervasive integration of digital media into daily life is reshaping how individuals encounter and internalize harmful contents. Unrestricted access exposes students to emotionally disruptive materials, including depictions of violence, substance use, and harassment, raising concerns about its impact on well-being. This study examines a serial mediation model linking antisocial media exposure to subjective well-being (SWB) through school-based victimization as sequential pathways. Using data from 326 high school students in Nigeria, we applied partial least squares structural equation modeling to test hypothesized relationships. Results indicate that antisocial media exposure was not directly associated with SWB but significantly predicted experiences of teacher and peer victimization. Peer victimization mediated the relationship between antisocial media exposure and SWB ($\beta = -0.023$, 95% CI: [$-0.054$, $-0.004$], $p < 0.05$). Furthermore, antisocial media exposure increased the likelihood of teacher victimization, which facilitated peer victimization, ultimately compromising SWB ($\beta = -0.030$, 95% CI: [$-0.058$, $-0.011$], $p < 0.05$). Effects varied by gender and academic level, underscoring intersectional risks linked to media exposure. Findings highlight the need for targeted interventions addressing both teacher and peer victimization in resource-constrained educational contexts.

## Impact statement

In today's digital age, students in schools have more access to mobile devices and are frequently exposed to harmful online content, such as violence, bullying and substance use, raising concerns about its impact on their well-being. Students' access to mobile devices presents a severe problem of antisocial media exposure, and while viewing antisocial content online may not directly endanger their wellness, it could promote bullying and mistreatment within school environments, particularly from teachers and peers. Experiences of bullying from these ecological sources are associated with reduced happiness, physical discomfort and psychological distress, suggesting that the negative effects of harmful online content often manifest in school relationships. When students resort to accessing antisocial media, teacher victimization may precede peer bullying, and together they significantly erode students' sense of wellness. Moreover, the impact of toxic online exposure varies by gender and academic level, indicating that some groups may be more vulnerable than others. These findings underscore the urgent need for schools, especially in resource-limited settings, to implement effective anti-bullying strategies and provide targeted support for students affected by toxic online environments.

## Introduction

In 2024, Nigeria recorded 36.7 million social media users, with internet penetration at 45.5% and mobile connectivity reaching 90.7% of the population (DataReportal, 2024). The ubiquity of digital media has heightened exposure to harmful content, particularly among vulnerable groups like students (Bandura, 2002; Perse, 2021). This risk is exacerbated in educational settings, where structural deficits, such as overcrowded classrooms and pupil–teacher ratios exceeding 80:1 in public schools, further undermine protective mechanisms and nuturing environment for students (UBEC, 2023). Situated within Bronfenbrenner's exosystem, the digitization of everyday life has normalized frequent and largely unregulated exposure to antisocial media through computer-mediated communication such as mobile devices and connectivity (Perse, 2021). Such exposure often includes depictions of firearms, graphic violence, substance use, hate speech,

harassment and sexualized imagery (Bandura, 2002; den Hamer et al., 2017; Bolaji et al., 2024). Collectively termed "antisocial media exposure," these encounters can precipitate significant emotional and behavioral disturbances, particularly when internalized (Calvete et al., 2020; Perse, 2021). Repeated exposure to antisocial media contents compounds emotional and behavioral disturbances, alter behaviors and pose substantial risks to well-being.

Empirical research from non-African contexts highlights the harmful influence of exposure to toxic digital media on subjective well-being (SWB), particularly in relation to online hate content (Keipi et al., 2018). Although digital platforms are ostensibly designed to promote connection and self-expression (Kim et al., 2016), exposure to anti-social content such as war-related or violent content can trigger fear, anxiety and depressive symptoms, that undermine mental health (Abu-Elenin et al., 2025). These challenges are especially pronounced in school settings, where the ripple effects of students' exposure to toxic media content could exacerbate antisocial behaviors (i.e., vandalism, deviance and aggression), promote poor peer to peer relationships, and stressors within the school environment. A growing body of literature indicates that harmful digital content often resurfaces in school-based interactions, increasing the risk of victimization through peer aggression and teacher hostility (Hong and Espelage, 2012; Boccio and Leal, 2022). These forms of victimization amplify the negative impact of antisocial media exposure, contributing to higher levels of depression, anxiety and cognitive dysfunction (Hopf et al., 2008; Hong and Espelage, 2012; Parris et al., 2022). For instance, social media content and the associated individual rumination are linked to exposure to cybervictimization, bullying and distress (Parris et al., 2022). Any typology of violence emanating from such victimization experience is interconnected with teacher unfairness, peer victimization and compromised life satisfactions, including school work-related anxiety (Huang, 2020). However, evidence remains nascent on how exposure to antisocial media content could channel school-based victimization and compromise the SWB of the students (Huang, 2020; Parris et al., 2022). A nuanced understanding of how digital and interpersonal ecological forces interact is essential for explaining the diverse developmental outcomes among students, particularly in resource-limited settings. By situating SWB within these dual ecological contexts (i.e., digital media and school environment), the present study investigates the combined influence of antisocial media exposure and school-based victimization on SWB, with attention to gender and academic-level differences among students in Nigeria.

## Theoretical framework

Bronfenbrenner's socioecological framework offers a compelling theoretical lens to interrogate these intersecting vulnerabilities of antisocial media exposure and SWB through school-based victimization. According to this framework, individual development is shaped by dynamic interactions across nested socio-environmental systems, from immediate interpersonal contexts to broader sociocultural structures (Bronfenbrenner, 1986). Within this framework, antisocial media exposure is situated within the digital environment of exosystem, a distal context that influences individuals *indirectly* through media consumption and how they socialize within the environment, while school-based victimization resides within the microsystem, where *direct* interpersonal interactions exert immediate developmental outcomes and consequences for individuals (Hong and Espelage, 2012; Sabri et al., 2013). The interplay between

these ecological layers may produce cumulative victimization effects, which could erode the ability to maintain positive SWB. For instance, repeated exposure to antisocial media content could destabilize interpersonal relationships such as student-teacher and/or student-student and exacerbate exposure to victimization and reinforce negative experiences that are consequential to well-being. Within Bronfenbrenner's framework, the family context, including parental education and living arrangements, constitutes a core element of the microsystem, directly shaping the cognitive, emotional and social development of individuals. Families serve as the primary socializing agents, influencing how young people interpret and respond to external stressors, such as exposure to antisocial media and school victimization (Bronfenbrenner, 1986; Smetana, 2011). Thus, variations in family structure and parental educational attainment may either buffer or amplify the adverse effects of these distal (e.g., antisocial media exposure) and proximal (e.g., school-based victimization) risks on SWB.

### Antisocial media exposure and SWB

Antisocial media exposure, particularly contents that are violent, sexually explicit or hate-driven, has been increasingly linked to outcomes, including elevated aggression, depressive symptoms and physiological stress reactions (Calvete et al., 2020; Perse, 2021; Abu-Elenin et al., 2025). Bronfenbrenner's ecological systems theory provides a conceptual understanding for these effects, emphasizing the reciprocal interactive relationship between individuals and their multilayered environments (Bronfenbrenner, 1995; Bronfenbrenner and Morris, 1998). Digital media operates as a distinct digital environment within this ecological model, particularly within the exosystem, where indirect influences can shape emotional and psychological development. Features such as instant access, user engagement, and algorithmic amplification facilitate exposure to antisocial media content, such as bullying, hate speech, and violent imagery, by prioritizing sensational or harmful posts in feeds and enabling rapid sharing across networks (Hong and Espelage, 2012). This negative contents are more harmful especially during the younger age, a developmental stage marked by identity formation and emotional sensitivity. Many young people lack the self-awareness needed to understand how antisocial media exposure affects their emotions and relationships, which can undermine their overall well-being (Fuld et al., 2009; Abu-Elenin et al., 2025). Although users retain some control over the content they engage with, ongoing exposure to antisocial media can lead to emotional numbness, increased anxiety and distorted views of reality (Fitzpatrick, 2016). As a result, the combined impact of ecological dynamics and technological design makes antisocial media exposure a serious concern for young population like students. Research has shown that such exposure can lead to nightmares, fear responses and aggressive behavior, all of which pose substantial risks to SWB (Keipi et al., 2018; Abu-Elenin et al., 2025).

### Antisocial media exposure and school-based victimization

Antisocial media exposure and school-based victimization represent the convergence of two ecological domains, such as digital and school ecologies, that interact synergistically and carry significant developmental consequences (Hong and Espelage, 2012). Antisocial media content, including depictions of violence, substance use, hate speech and bullying victimization, functions as a digital environment within Bronfenbrenner's exosystem, indirectly shaping behavioral norms and influencing how students navigate school

settings and are perceived by others (Wong et al., 2008). These influences may manifest in behaviors, such as social withdrawal, hypervigilance or reactive aggression, which can provoke peer rejection or bullying victimization (Longobardi et al., 2020). At the same time, educators working within rigid institutional structures may misinterpret these behaviors as defiance or misconduct, responding with punitive or dismissive actions (Wong et al., 2008). The reciprocal nature of peer and teacher responses can reinforce victimization, as teacher rejection may legitimize peer aggression, while peer hostility may elicit behaviors that further alienate students from educators (Bronfenbrenner and Morris, 1998; Boccio and Leal, 2022). Although prior research has examined the behavioral effects of antisocial media exposure, it has not sufficiently addressed how such exposure contributes to dual victimization from both peers and teachers, particularly among individuals who internalize negative digital content (Parris et al., 2022). Antisocial media exposure should therefore be understood not as a peripheral influence, but as a consequential ecological force that actively reshapes school-based interactions and increases vulnerability to interpersonal harm (Parris et al., 2022).

## Serial pathway through school-based victimization

Exposure to antisocial digital content may impair emotional regulation and foster problematic internet use as a maladaptive coping strategy (Gioia et al., 2021). The maladaptive response may manifest as school-based behavioral dysregulation that elicits negative responses from teachers and peers. Perceived teacher unfairness and peer victimization, whether independently or cumulatively predict diminished life satisfaction, psychological distress and a reduced sense of safety (Gini et al., 2018). Peer victimization is closely tied to social rejection and low self-worth, reinforcing the marginalization that follows teacher misjudgment and escalating the erosion of well-being (Boulton and Smith, 1994). The internalized effects of antisocial media content often surface in school environments as behavioral cues, such as irritability, withdrawal or perceived defiance, that are frequently misinterpreted by educators, particularly in emotionally unsupportive classrooms (Longobardi et al., 2022). Students who experienced bullying victimization report feeling unsafe and less connected at school, reinforcing the link between experiecning aggressive behavior and diminished SWB (Goldweber et al., 2013).

Digital media is a critical element in the socialization process, and when it contains hostile and damaging content, it creates an adverse socialization experience for individuals, which creates the likelihood of shaping behavioral tendencies that are easily misunderstood in structured institutional settings (Perse, 2021). Hostile social interactions often begin with subtle acts of exclusion or ridicule. In today's digital age, students' use of social media platforms can rapidly escalate negative encounters into persistent, public and psychologically damaging exposure, such as cyberbullying victimization that could embed aggression into the fabric of school socialization (Hong and Espelage, 2012). Teacher–student conflict is a critical early signal, as it may not only erode students' sense of safety but also serve as a social cue that legitimizes peer rejection and victimization, making it a likely starting point for broader student-to-student conflict (ten Bokkel et al., 2021). Students who experience conflictual relationships with teachers are more likely to be disliked by peers and subsequently targeted for bullying victimization, especially in classrooms where teachers are otherwise perceived as highly responsive to others (Longobardi et al., 2020). Empirical evidence further shows that peer

victimization and perceived teacher unfairness significantly undermine adolescents' life satisfaction, with a diminished sense of belonging to school serving as a key mediating mechanism through which these negative social experiences exert their psychological toll (Huang, 2020).

Exposure to antisocial media can distort students' emotional expressions and social behaviors, leading teachers to misinterpret these cues and respond with punitive or emotionally neglectful actions. Such misjudgments may result in teacher victimization, including harsh disciplinary practices, public reprimands or emotional withdrawal, all of which erode students' mental health and sense of belonging (ten Bokkel et al., 2021). These negative teacher responses can act as social signals to peers, implicitly legitimizing the marginalization of the targeted student and increasing the likelihood of peer rejection and bullying victimization (Longobardi et al., 2020). Students' perceptions of teachers' moral authority and their attitudes toward bullying victimization significantly influence bullying, suggesting that when teachers fail to intervene effectively or are perceived as indifferent, they may inadvertently legitimize aggression, setting the stage for peer victimization to follow (Lee, 2010). Peer victimization significantly undermines adolescents' life satisfaction, primarily through reduced school belonging and heightened anxiety, effects that are often rooted in earlier relational breakdowns with teachers (Huang, 2020). This cascading pattern suggests that antisocial media exposure contributes to a serial process of victimization, beginning with teacher–student conflict and culminating in peer bullying, ultimately compromising the SWB.

## Current study

This study interrogates the direct and indirect effects of antisocial media exposure on school-based victimization, specifically peer and teacher bullying victimization, and its cascading impact on the SWB of students in Nigeria. Employing a serial mediation model, the research examines how antisocial media exposure contributes to poor well-being through the compounded effects of teacher and peer victimization.

Additionally, this study examines how gender and academic level shape the relationships between antisocial media exposure, school victimization (including peer- and teacher-bullying victimization) and mental health outcomes, offering an intersectional lens on vulnerability within educational settings. Gendered patterns in digital media use and emotional distress are well-documented: males are more likely to engage in online aggression and be perceived as aggressors, while females often seek social validation through digital platforms, increasing their susceptibility to peer rejection and emotional harm (Ludeman, 2004; Andreassen, 2015; Brown and Tiggemann, 2016). Gender diverse youth show the highest levels of media engagement and depressive symptoms, reflecting both heightened vulnerability and deeper reliance on digital spaces for identity affirmation (Klinger et al., 2024). Moreover, cyberbullying has been shown to undermine the protective effects of social media on self-esteem and support, with male and female students exhibiting distinct psychological responses (Zhang et al., 2023). In this context, males frequently emerge as perpetrators of media-based aggression, whereas females are disproportionately positioned as victims (Reed et al., 2016). Male and female observers tended to project positive stereotypes onto same-gender faces, with male faces more often perceived as aggressors and female faces as victims (Bracci et al., 2021). Within school environments, male-to-male bullying victimization is prevalent, and gendered power asymmetries often result in boys targeting girls (Hong and Espelage, 2012). Moreover, male exposure to violent

digital content has been associated with increased real-world aggression and emotional desensitization (Hopf et al., 2008). This evidence suggests that gender differences could emerge from the pathways from antisocial media exposure to school-based victimization and well-being.

Despite growing evidence on age-related differences in bullying victimization and cyberbullying, there remains a paucity of research examining how academic level influences the interplay between antisocial media exposure, school victimization and psychological well-being, particularly within Nigerian educational contexts. Academic level may shape both vulnerability and coping capacities, yet its role in these dynamics is underexplored. Students transitioning between educational stages face heightened bullying victimization risks due to shifting peer norms and identity pressures, with younger students especially vulnerable to exclusion when they fail to conform to dominant youth cultures (Moody and Stahel, 2025). Empirical evidence further shows that younger adolescents are more likely to experience school bullying victimization, while older students engage more in cyberbullying and exhibit complex bully-victim profiles, underscoring the need for age-sensitive prevention strategies (Pichel et al., 2021). These developmental differences suggest that lower-level students may be more susceptible to media-induced harm, whereas higher-level students may encounter subtler forms of victimization or possess greater resilience. By investigating these group differences, the study contributes to a more nuanced understanding of how intersecting social dimensions condition students' exposure to and experiences of digital aggression and school-based victimization.

## Method

### Study design, participants and data collection

This study adopted a cross-sectional design to recruit 326 students from three high schools in Nigeria. Data collection was conducted over a 5-month period, from January to May 2024. Participants were selected using a purposive sampling technique to ensure representation across the targeted educational levels. All students provided informed consent to parents before participation. Ethical approval was obtained from the heads of each participating school. Data were collected using a validated, self-reported questionnaire designed to assess the constructs of interest. A priori power analysis for structural equation modeling (SEM) was considered, assuming a medium effect size (Cohen's $f^2 = 0.30$), an alpha level of 0.05 and a desired statistical power of 0.80. With a sample size of 326, the study had sufficient power to estimate a model comprising four latent variables and ~20–30 parameters. This sample size was adequate to ensure reliable parameter estimation and acceptable model fit indices.

### Measures

#### Subjective well-being

SWB was conceptualized as a multidimensional construct reflecting individuals' self-perceived quality of life, encompassing emotional, physical and psychological dimensions. This approach aligns with contemporary perspectives that define SWB not merely as the absence of distress but as the presence of positive functioning and life satisfaction (Das et al., 2020). In this study, SWB was operationalized through three interrelated components: subjective happiness, subjective health and psychological health, each serving as a reflective indicator of the latent construct. Subjective happiness was measured using the four-item *Subjective Happiness Scale* (Lyubomirsky and Lepper, 1999), which captures individuals' global assessment of their happiness relative to peers and their general life satisfaction. Items were rated on a 7-point Likert scale, with higher scores indicating greater perceived happiness. This subscale demonstrated acceptable internal consistency ($\alpha = 0.703$), and the standardized factor loading was 0.52, supporting its contribution to the overall SWB construct. Subjective health was assessed using two items adapted from prior research focusing on self-rated health and comparative health perception (Cislaghi and Cislaghi, 2019). Participants responded to statements such as "I feel satisfied with my overall health" and "My health is better compared to others" on a 5-point scale ranging from 1 (*Not at all*) to 5 (*Completely*). These items captured both absolute and relative health perceptions, with a factor loading of 0.61, indicating a moderate contribution to the latent construct. Psychological Health was measured using six items derived from the psychological domain of the WHO Quality of Life instrument (WHOQOL, 1998). This subscale assessed both positive psychological states (e.g., life satisfaction and body image) and the absence of negative emotions (e.g., anxiety and depression). Responses were recorded on a 5-point Likert scale from 1 (*Strongly disagree*) to 5 (*Strongly agree*), with higher scores reflecting better psychological well-being. This dimension showed strong internal consistency ($\alpha = 0.79$) and a factor loading of 0.71. All three subscales were modeled as first-order indicators of a second-order latent construct "Subjective Well-Being" within the structural equation model. The measurement model demonstrated satisfactory convergent validity, with all factor loadings exceeding the minimum threshold of 0.40. Discriminant validity was confirmed using the Heterotrait–Monotrait (HTMT) ratio, with all values falling below the recommended cutoff of 0.85. This multidimensional operationalization allowed for a robust and theoretically grounded assessment of SWB among students in the study.

### Antisocial media exposure

This construct was assessed using the antisocial content subscale of the Content-based Media Exposure Scale, a validated tool designed to evaluate adolescents' exposure to harmful media content (den Hamer et al., 2017). This subscale includes eight items that ask participants how frequently they encounter media portraying behaviors, such as physical aggression, substance use, criminal acts, and bullying victimization. Responses were recorded on a 5-point Likert scale ranging from 1 (*Never*) to 5 (*Very often*). Example items include exposure to scenes of people fighting, using drugs or engaging in threatening behavior. These items were treated as indicators of a latent variable representing antisocial media exposure in the structural equation model. The items in the scale show strong factor loadings (0.50–0.64).

### Teacher victimization

Teacher victimization was assessed using a six-item scale developed to capture students' experiences of mistreatment by teachers, adapted from a version of the California School Climate and Safety Survey (Chen and Wei, 2011). This scale was validated by teachers who participated in data collection, as assessed for its cultural relevance. Participants were asked to indicate how often they had experienced specific negative behaviors from teachers, using a 5-point Likert scale ranging from 1 (*Never*) to 5 (*Very often*). The

items included: "Teachers hit, beat or slapped you," "Teachers cursed you," "Teachers ignored you purposely," "Teachers touched you unnecessarily," "Teachers singled you out in class for verbal abuse" and "Teachers screamed at you in class." These items were modeled as indicators of a single latent construct, "Teacher Victimization," in the structural equation model. Higher scores reflected more frequent experiences of teacher-perpetrated victimization. The scale demonstrated acceptable internal consistency and was appropriate for capturing the multidimensional nature of teacher–student conflict in the school context, with factor loading ranging from 0.67 to 9.78.

### Peer victimization

Peer victimization was measured using the Multidimensional Peer Victimization Scale developed by Mynard and Joseph (2000). This 16-item instrument assesses adolescents' experiences of victimization by peers across four dimensions: physical victimization, verbal victimization, social manipulation and attacks on property. Each subscale consists of four items, and participants responded using a 3-point Likert scale: 0 (*Not at all*), 1 (*Once*) and 2 (*More than once*). Sample items include "A student has hit or kicked me" (physical), "A student has called me names" (verbal), "A student has tried to turn my friends against me" (social) and "A student has taken or damaged something of mine" (property). In this study, the four subscales were modeled as indicators of a single latent construct, "Peer Victimization," within the structural equation model. This scale was validated by teachers who participated in data collection, as assessed for its cultural relevance. This approach allowed for a comprehensive representation of peer victimization experiences, and the scale demonstrated good internal consistency and construct validity in the sample.

### Sociodemographic/control variables

Participants' background, such as age, was grouped into categories ranging from 13 to 22 years. Gender was recorded as either male or female. Academic level was measured by a lower academic level (SS1) or an upper academic level (SS2). Living arrangement was assessed by asking whether students lived with both parents, one parent or with a foster family or grandparents. Academic performance was calculated by averaging students' scores across all subjects for the academic year, and then grouped into three categories: <50%, between 50 and 69% and ≥70%. Parental education was measured separately for mothers and fathers, using seven categories: illiterate, primary school, middle school, high school, post-secondary diploma, graduate/postgraduate and professional degree (see Table 1).

### Analysis

The data analysis was conducted in multiple stages. First, descriptive statistics were computed using SPSS Version 25.0 to summarize the sociodemographic characteristics and the distribution of observed and latent variables. This included frequencies, means (*M*) and standard deviations (SDs). Bivariate correlations were then examined to explore the relationships among study variables and assess their suitability for inclusion in the structural model. Partial least square SEM was conducted using Smart Pls 4, by iteratively applying ordinary least square (OLS) regressions to maximize the explained variance ($R^2$) of the endogenous constructs. Convergent validity was assessed by examining factor loadings, with factor

**Table 1.** Demographic characteristics of study participants, *N* = 326

| Variables | | Frequency (%) | *M* ± SD |
|---|---|---|---|
| Age | 13 | 2 (0.6) | 16.16 ± 1.21 |
| | 14 | 19 (5.8) | |
| | 15 | 82 (25.2) | |
| | 16 | 95 (29.1) | |
| | 17 | 87 (26.7) | |
| | 18 | 35 (10.7) | |
| | 19–22 | 6 (1.8) | |
| Gender | Female | 224 (68.7) | 0.31 ± 0.46 |
| | Male | 102 (31.3) | |
| Academic level | Lower academic level (SS1) | 159 (48.8) | 0.51 ± 0.50 |
| | Upper academic level (SS2) | 167 (51.2) | |
| Living arrangement | Living with foster family/grandparents | 23 (7.1) | 0.94 ± 0.83 |
| | Living with mother or father | 56 (14.1) | |
| | Living with both parents | 247 (75.8) | |
| Academic performance | <50% | 24 (7.4) | 1.32 ± 0.61 |
| | 50–69% | 173 (53.1) | |
| | ≥70% | 129 (39.5) | |
| Father's education | Illiterate | 2 (0.6) | 3.80 ± 1.46 |
| | Primary school | 10 (3.1) | |
| | Middle school | 48 (14.7) | |
| | High school | 109 (33.4) | |
| | Intermediate/Post high school diploma | 31 (9.5) | |
| | Graduate/Postgraduate | 73 (22.4) | |
| | Professional degree | 53 (16.3) | |
| Mother's education | Illiterate | 4 (1.2) | 3.66 ± 1.50 |
| | Primary school | 14 (4.3) | |
| | Middle school | 51 (15.6) | |
| | High school | 116 (35.6) | |
| | Intermediate/Post high school diploma | 26 (8.0) | |
| | Graduate/Postgraduate | 66 (20.2) | |
| | Professional degree | 49 (15.0) | |

M, mean; SD, standard deviation.

loadings of 0.50 or higher considered acceptable. Discriminant validity was evaluated using the HTMT ratio of correlations, with values below 0.85 indicating adequate discriminant validity. The analysis resolved all multicollinearity issues using Variance inflation factors (VIF) a statistical measure in regression analysis that detects multicollinearity (high correlation between independent variables), indicating how much the variance of a coefficient is inflated, with values above 5 or 10 suggesting problematic correlations that can make model results unreliable. To test for mediation

effects, a bootstrapping procedure with 2,000 resamples was used to estimate indirect effects and generate bias-corrected confidence intervals (CIs). This approach provided robust estimates of standard errors and significance levels for indirect paths.

A multigroup analysis was conducted to examine potential differences in the structural model across gender (male vs. female) and academic level (lower academic level vs. upper academic level). Multigroup analysis was conducted to examine whether the structural relationships in the model differed significantly across gender and academic levels, thereby enhancing the contextual sensitivity of the findings. Measurement invariance was assessed to ensure that the constructs were interpreted equivalently across groups, supporting the validity and robustness of the group comparisons. Effect sizes were interpreted using standardized regression coefficients ($\beta$), with values of 0.10, 0.30 and 0.50 representing small, medium and large effects, respectively. The explanatory power of the model was assessed using $R^2$ values, where 0.25, 0.50 and 0.75 indicated low, moderate and high levels of explained variance.

## Result

The study sample shown in Table 1 comprised 326 students enrolled in lower academic level and upper academic level in Nigeria, with a mean age of 16.16 years (SD = 1.21). The majority of participants were female (68.7%), and most reported living with both parents (75.8%). In terms of academic performance, over half of the students (53.1%) reported average scores between 50 and 69%, and parental education levels varied, with a substantial proportion of fathers (33.4%) and mothers (35.6%) having completed high school.

Table 2 results show that SWB was negatively correlated with peer victimization ($r = -.25$, $p < .01$) and teacher victimization ($r = -.17$, $p < .01$). Peer victimization was strongly correlated with teacher victimization ($r = .56$, $p < .01$), and positively associated with antisocial media exposure ($r = .12$, $p < .05$). Teacher victimization was negatively associated with academic level ($r = -.13$, $p < .05$) and living arrangement ($r = -.15$, $p < .01$). Antisocial media exposure was positively correlated with gender ($r = .17$, $p < .01$) and negatively with age group ($r = -.16$, $p < .01$).

## Factor loading, convergent validity and average variance ext

Table 3 presents the results of a factor analysis, showing how well individual items load onto their respective latent constructs (e.g., "subjective well-being" and "antisocial media exposure"), with reliability indicators confirming internal consistency. Higher factor loadings (closer to 1) indicate stronger relationships between items and their constructs, while $\alpha$, convergent validity and average variance extracted values suggest good reliability of the constructs.

Table 4 shows the HTMT ratios used to assess discriminant validity, which tests whether constructs in a model are truly distinct from one another. All HTMT values are below the common threshold of 0.85, indicating that the constructs are sufficiently different from each other.

## Path analysis

The path analysis in Table 5 reveals that antisocial media exposure is significantly associated with teacher victimization ($\beta = 0.293$, $p < 0.001$) and peer victimization ($\beta = 0.108$, $p < 0.05$). However, the direct impact of antisocial media exposure on SWB is not statistically significant. Peer victimization ($\beta = -0.210$, $p < 0.01$) and teacher victimization ($\beta = -0.123$, $p < 0.05$) are negatively associated with SWB and Tea. Teacher victimization is positively associated with peer victimization ($\beta = 0.490$, $p < 0.001$), indicating a reinforcing dynamic between these forms of victimization. Demographic variables, such as age and parental education, show no significant direct effects on either school-based victimization or well-being. However, living arrangement is negatively associated with peer victimization ($\beta = -0.090$, $p < 0.05$) and teacher victimization ($\beta = -0.113$, $p < 0.01$). All VIF (variance inflation factors) values are below 2.1, indicating no multicollinearity concerns.

The $R^2$ values in Figure 1 reflect the proportion of variance explained by the predictors in the model. Teacher victimization ($R^2 = 0.096$) indicates that 9.6% of its variance is explained by antisocial media exposure and other factors. Peer victimization ($R^2 = 0.307$) shows a stronger explanatory power, likely due to contributions from both antisocial media exposure and teacher victimization. SWB ($R^2 = 0.085$) has the lowest explained variance, suggesting that additional unmeasured factors may significantly influence this outcome.

**Table 2.** Correlation matrix

| | 1 | 2 | 3 | 4 | 5 | 6 | 7 | 8 | 9 | 10 | 11 |
|---|---|---|---|---|---|---|---|---|---|---|---|
| 1. Subjective well-being | 1 | −0.249** | −0.167** | 0.052 | −0.090 | −0.035 | 0.126* | 0.061 | 0.091 | 0.047 | 0.011 |
| 2. Peer victimization | | 1 | 0.563** | 0.009 | 0.123* | −0.006 | −0.044 | −0.182** | 0.031 | 0.024 | −0.040 |
| 3. Teacher victimization | | | 1 | −0.024 | 0.099 | −0.042 | −0.130* | −0.151** | −0.053 | −0.008 | −0.033 |
| 4. Antisocial media exposure | | | | 1 | 0.174** | −0.159** | 0.034 | 0.106 | 0.094 | 0.107 | 0.050 |
| 5. Gender | | | | | 1 | −0.106 | 0.063 | 0.085 | 0.177** | 0.299** | 0.349** |
| 6. Age group | | | | | | 1 | 0.169** | 0.031 | −0.041 | −0.069 | −0.112* |
| 7. Academic level | | | | | | | 1 | 0.027 | −0.170** | −0.026 | −0.052 |
| 8. Living Arrangement | | | | | | | | 1 | 0.216** | 0.042 | 0.032 |
| 9. Academic Performance | | | | | | | | | 1 | 0.224** | 0.190** |
| 10. Fathers' education | | | | | | | | | | 1 | 0.711** |
| 11. Mothers' education | | | | | | | | | | | 1 |

*Note*: *$p < 0.05$, **$p < 0.01$.

**Table 3.** Factor loading, convergent validity and AVE

|  | Factor loadings | α | CR | AVE |
|---|---|---|---|---|
| Subjective well-being |  | 0.739 | 0.893 | 0.658 |
| Happiness | 0.907 |  |  |  |
| Subjective health | 0.531 |  |  |  |
| Psychological health | 0.933 |  |  |  |
| Antisocial media exposure |  | 0.760 | 0.784 | 0.401 |
| SME8 | 0.613 |  |  |  |
| SME7 | 0.706 |  |  |  |
| SME6 | 0.604 |  |  |  |
| SME5 | 0.626 |  |  |  |
| SME4 | 0.669 |  |  |  |
| SME3 | 0.602 |  |  |  |
| SME2 | 0.605 |  |  |  |
| Teacher victimization |  | 0.850 | 0.855 | 0.8625 |
| TVict2 | 0.785 |  |  |  |
| TVict3 | 0.832 |  |  |  |
| TVict4 | 0.781 |  |  |  |
| TVict5 | 0.809 |  |  |  |
| TVict6 | 0.744 |  |  |  |
| Peer victimization |  | 0.899 | 0.907 | 0.768 |
| Verbal V | 0.821 |  |  |  |
| Social V | 0.923 |  |  |  |
| Property V | 0.903 |  |  |  |
| Physical V | 0.854 |  |  |  |

CR, convergent validity.

**Table 4.** Discriminant validity (Heterotrait–Monotrait [HTMT])

|  | 1 | 2 | 3 | 4 |
|---|---|---|---|---|
| 1. Subjective well-being | - |  |  |  |
| 2. Antisocial media exposure | 0.086 | - |  |  |
| 3. Peer victimization | 0.274 | 0.305 | - |  |
| 4. Teacher victimization | 0.254 | 0.345 | 0.607 | - |

### Indirect pathways

The mediation analysis in Table 6 highlights how antisocial media exposure indirectly affects SWB through school-based victimization pathways. The indirect effect via peer victimization is statistically significant ($\beta = -0.023$, 95% CI: $[-0.054–0.004]$, $p < 0.05$), suggesting a modest but meaningful negative impact. The path through teacher victimization alone is not significant ($\beta = -0.036$, 95% CI: $[-0.072, 0.001]$, $p = 0.116$), indicating that teacher victimization by itself may not mediate the relationship. The combined pathway from antisocial media exposure to teacher victimization to peer victimization and then to poor well-being is significant ($\beta = -0.030$, 95% CI: $[-0.058, -0.011]$, $p < 0.05$), underscoring a more complex and impactful mediation route. These findings suggest that peer victimization plays a critical role in translating media exposure into diminished well-being, especially when preceded by teacher victimization.

**Table 5.** Standardized beta coefficients in the direct paths

|  | β | VIF | SE | t test |
|---|---|---|---|---|
| Antisocial media exposure -> Subjective well-being | 0.069 | 1.151 | 0.075 | 0.921 |
| Antisocial media exposure -> Peer victimization | 0.108* | 1.134 | 0.049 | 2.226 |
| Antisocial media exposure -> Teacher victimization | 0.293*** | 1.038 | 0.072 | 4.043 |
| Age -> Subjective well-being | −0.054 | 1.016 | 0.070 | 0.764 |
| Age -> Peer victimization | 0.010 | 1.016 | 0.043 | 0.241 |
| Age -> Teacher victimization | −0.039 | 1.014 | 0.059 | 0.672 |
| Father's education -> Subjective well-being | 0.089 | 2.044 | 0.081 | 1.101 |
| Father's education -> Peer victimization | 0.087 | 2.033 | 0.082 | 1.064 |
| Father's education -> Teacher victimization | 0.003 | 2.033 | 0.083 | 0.032 |
| Living arrangement -> Subjective well-being | 0.018 | 1.052 | 0.065 | 0.269 |
| Living arrangement -> Peer victimization | −0.090* | 1.041 | 0.051 | 1.776 |
| Living arrangement -> Teacher victimization | −0.113* | 1.026 | 0.061 | 1.852 |
| Mother's education -> Subjective well-being | −0.064 | 2.061 | 0.080 | 0.794 |
| Mother's education -> Peer victimization | −0.110 | 2.044 | 0.068 | 1.620 |
| Mother's education -> Teacher victimization | −0.025 | 2.043 | 0.077 | 0.322 |
| Peer victimization -> Subjective well-being | −0.210** | 1.460 | 0.073 | 2.860 |
| Teacher victimization -> Subjective well-being | −0.123* | 1.473 | 0.075 | 1.648 |
| Teacher victimization -> Peer victimization | 0.490*** | 1.122 | 0.064 | 7.605 |

*Note*: *$p < 0.05$, **$p < 0.01$, ***$p < 0.001$; $\beta$, standardized beta; SE, standard error.

### Gender and academic level variations

Among males, antisocial media exposure was significantly associated with increased teacher victimization ($\beta = 0.458$, $p < 0.05$); father's education positively predicted SWB ($\beta = 0.349$, $p < 0.05$); and mother's education negatively predicted SWB ($\beta = -0.256$, $p < 0.05$). Meanwhile, among females, antisocial media exposure was significantly associated with increased peer victimization ($\beta = 0.100$, $p < 0.05$); living arrangement negatively predicted peer victimization ($\beta = -0.079$, $p < 0.05$); and peer victimization was strongly associated with lower SWB ($\beta = -0.340$, $p < 0.001$), showing a divergence in how antisocial media exposure and family factors relate to well-being across genders. In contrast, teacher victimization significantly predicted peer victimization in both males ($\beta = 0.393$, $p < 0.001$) and females ($\beta = 0.571$, $p < 0.001$), indicating convergence in this pathway. Among lower-level students, antisocial media exposure was significantly associated with increased teacher victimization ($\beta = 0.312$, $p < 0.01$); age negatively predicted SWB ($\beta = -0.158$, $p < 0.05$); living arrangement negatively predicted both peer ($\beta = -0.198$, $p < 0.05$) and teacher

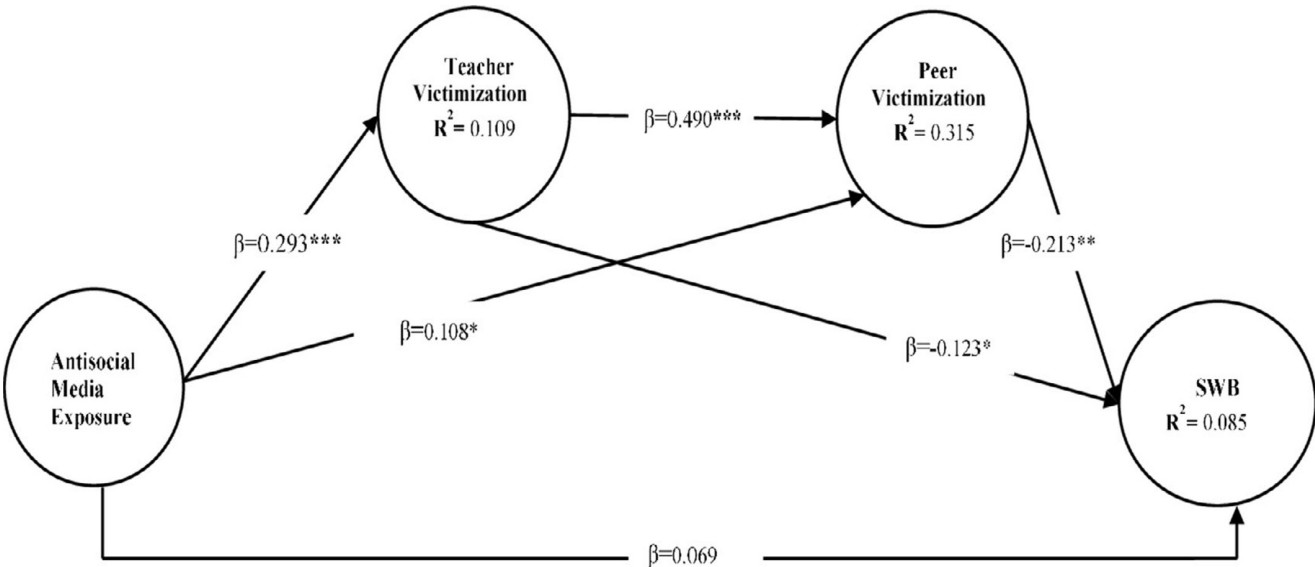

**Figure 1.** The structural equation of final model. *p < 0.05, **p < 0.01 and ***p < 0.001. SWB, subjective well-being; controlled for age, living arrangement, father's education and mother's education.

**Table 6.** Indirect (mediation) effects

| Indirect effects | β | LB-UB (95% CI) |
|---|---|---|
| Antisocial media exposure -- > Peer victimization -- > SWB | −0. 023* | (−0.054; −0.004) |
| Antisocial media exposure -- > Teacher victimization -- > SWB | −0.036 | (−0.073; 0.001) |
| Antisocial media exposure -- > Teacher victimization -- > Peer victimization -- > SWB | −0.030* | (−0.058; −0.011) |

*Note*: *p < 0.05, **p < 0.01, ***p < 0.001; β, standardized beta; CI, confidence interval; LB, lower bound; UB, upper bound.

victimization ($\beta = -0.171$, $p < 0.05$); and peer victimization was negatively associated with SWB ($\beta = -0.247$, $p < 0.05$). Among upper-level students, antisocial media exposure was also significantly associated with increased teacher victimization ($\beta = 0.227$, $p < 0.05$), and teacher victimization negatively predicted SWB ($\beta = -0.215$, $p < 0.05$), showing convergence in the antisocial media–teacher victimization link but divergence in how victimization impacts well-being across academic levels. Teacher victimization significantly predicted peer victimization in both lower-level ($\beta = 0.471$, $p < 0.001$) and upper-level students ($\beta = 0.494$, $p < 0.001$), indicating convergence in this pathway (Table 7).

## Discussion

This study examined the pathways from antisocial media exposure to SWB of students, through school-based victimization such as peer and teacher victimization. Moving beyond linear models that assume a direct link between antisocial media exposure and well-being, our findings indicate a sequential mediated pathway. Specifically, antisocial media exposure heightens the risk of teacher victimization, which subsequently facilitates peer bullying, culminating in reduced well-being. These results challenge assumptions that such antisocial media exposure operates in isolation, highlighting instead the relational and institutional

contexts through which its effects unfold (Keipi et al., 2018; Abu-Elenin et al., 2025).

The results established the relevance of Bronfenbrenner's ecological systems theory, which conceptualizes development as shaped by interactions between nested environmental systems (Bronfenbrenner and Morris, 1998). Within this framework, antisocial media exposure functions as a distal exosystemic force whose negative impact on well-being is mediated through proximal microsystemic interactions, namely those occurring within school environments. The inclusion of both peer and teacher victimization in the model provides a more holistic understanding of how digital harm is internalized and socially reinforced.

The significant association between antisocial media exposure and teacher victimization highlights an often-overlooked dynamic in educational settings. Antisocial media content may dysregulate students' emotional states and behavioral expressions. In rigid or emotionally unsupportive classrooms, such behaviors are frequently misinterpreted as defiance or misconduct, prompting punitive responses from educators. These disciplinary actions not only harm the targeted student but also signal to peers that the individual is socially devalued, thereby legitimizing further victimization (Longobardi et al., 2022). This cascading effect, where institutional rejection precipitates peer aggression, reinforces a cycle of marginalization and impaired well-being (Huang, 2020).

The serial mediation pathway identified in this study substantiates the theoretical claim that cumulative victimization across hierarchical relationships is a key mechanism through which antisocial media exposure influences SWB. Peer victimization, in particular, emerges as the most immediate and damaging predictor of reduced well-being, reflecting the acute social consequences of media exposure in adolescent peer contexts. Within the socioecological framework, antisocial media exposure initiates a chain of risk across interconnected systems, beginning with emotional dysregulation and culminating in relational exclusion and impaired well-being (Bronfenbrenner, 1995; Hong and Espelage, 2012).

Within these negative experiences among the students, family factors play a significant role in their experience of teacher and peer victimization. For instance, a more positive living arrangement of the student could cushion the possibility of encountering a

**Table 7.** Standardized beta coefficients and standard error based on gender and academic level

| | Gender | | | | Academic level | | | |
| | Male | | Female | | Lower-level students | | Upper-level students | |
| Direct paths | $\beta$ | SE | $\beta$ | SE | $\beta$ | SE | $\beta$ | SE |
|---|---|---|---|---|---|---|---|---|
| Antisocial media exposure - > Subjective well-being | 0.232 | 0.154 | −0.021 | 0.080 | 0.192 | 0.126 | −0.045 | 0.100 |
| Antisocial media exposure - > Peer victimization | 0.115 | 0.128 | 0.100* | 0.056 | 0.103 | 0.078 | 0.137 | 0.084 |
| Antisocial media exposure - > Teacher victimization | 0.458* | 0.125 | 0.118 | 0.082 | 0.312** | 0.100 | 0.227* | 0.086 |
| Age - > Subjective well-being | −0.150 | 0.109 | −0.054 | 0.070 | −0.158* | 0.077 | −0.017 | 0.097 |
| Age - > Peer victimization | 0.108 | 0.086 | −0.030 | 0.044 | 0.048 | 0.067 | −0.030 | 0.057 |
| Age - > Teacher victimization | −0.068 | 0.110 | −0.027 | 0.070 | 0.031 | 0.077 | −0.097 | 0.089 |
| Father's education - > Subjective well-being | 0.349* | 0.133 | 0.048 | 0.092 | 0.107 | 0.111 | 0.036 | 0.126 |
| Father's education - > Peer victimization | 0.143 | 0.165 | 0.051 | 0.090 | 0.161 | 0.099 | −0.015 | 0.125 |
| Father's education - > Teacher victimization | −0.006 | 0.142 | 0.051 | 0.106 | 0.023 | 0.132 | −0.035 | 0.105 |
| Living arrangement - > Subjective well-being | −0.042 | 0.140 | 0.054 | 0.067 | 0.078 | 0.099 | −0.028 | 0.081 |
| Living arrangement - > Peer victimization | −0.125 | 0.103 | −0.079* | 0.060 | −0.198* | 0.079 | 0.028 | 0.068 |
| Living arrangement - > Teacher victimization | −0.057 | 0.122 | −0.141 | 0.069 | −0.171* | 0.086 | −0.004 | 0.069 |
| Mother's education - > Subjective well-being | −0.256* | 0.128 | 0.002 | 0.085 | −0.017 | 0.107 | −0.062 | 0.128 |
| Mother's education - > Peer victimization | −0.120 | 0.130 | −0.113 | 0.080 | −0.115 | 0.098 | −0.095 | 0.091 |
| Mother's education - > Teacher victimization | 0.037 | 0.121 | −0.123 | 0.096 | −0.050 | 0.125 | −0.002 | 0.084 |
| Peer victimization - > Subjective well-being | 0.026 | 0.124 | −0.340*** | 0.077 | −0.247* | 0.114 | −0.111 | 0.105 |
| Teacher victimization - > Subjective well-being | −0.183 | 0.131 | −0.082 | 0.089 | −0.072 | 0.096 | −0.215* | 0.100 |
| Teacher victimization - > Peer victimization | 0.393*** | 0.143 | 0.571*** | 0.056 | 0.471*** | 0.079 | 0.494*** | 0.097 |

*Note:* *$p < 0.05$, **$p < 0.01$, ***$p < 0.001$; $\beta$, standardized beta; SE, standard error.

bothersome teacher and peer victimization. The proportion of variance explained in the model offers further insight into the mechanisms at play. While the model demonstrates strong explanatory power for peer victimization, its capacity to account for teacher victimization and SWB is more modest. This suggests that peer dynamics are especially sensitive to external cues, such as antisocial media exposure and institutional behavior, reinforcing the importance of relational processes within the school microsystem. In contrast, teacher behavior appears to be influenced by a broader constellation of factors, including institutional culture, professional training and implicit bias, that extend beyond the scope of antisocial media exposure. Based on the effect size of SWB of the student, the interpretation of the findings should take into consideration the unmeasured psychosocial construct that could influence the experience of the students within this theoretical framework. Thus, antisocial media exposure and school-based victimization, along with other socioeconomic conditions, accounted for a fraction of the SWB of the students.

The analysis also reveals important gender and academic-level differences in how toxic digital media and victimization dynamics affect well-being. Among male students, antisocial media exposure significantly predicts teacher victimization, suggesting that boys may be more vulnerable to misinterpretation and disciplinary action in response to media-influenced behaviors. Notably, paternal education emerged as a significant predictor of well-being among males, indicating that higher paternal educational attainment may serve as a protective factor, likely through its association with household income and psychosocial support (Walsh et al., 2024).

For female students, peer victimization was a strong and significant negative predictor of poor well-being, underscoring the emotional toll of relational aggression. This aligns with existing literature suggesting that girls are more likely to engage with digital media for social validation and are disproportionately targeted in peer-based victimization, often rooted in gendered power dynamics (Andreassen, 2015). When disaggregated by academic level, students in the lower academic level exhibited significant pathways from antisocial media exposure to teacher victimization and from peer victimization to reduced well-being, suggesting heightened vulnerability during this transitional academic stage. In the upper academic level, teacher victimization significantly predicted both peer victimization and reduced well-being, indicating a more entrenched pattern of cascading victimization in the final year of secondary school, possibly exacerbated by academic pressure and increased digital access. In sum, this study advances a nuanced, intersectional understanding of how antisocial media exposure interacts with institutional and relational dynamics in shaping the well-being of students. By situating digital harm within broader ecological and demographic contexts, it underscores the need for targeted interventions that address both teacher and peer victimization, particularly in resource-constrained educational settings.

Academic level emerged as a salient moderator in the relationship between toxic digital media, school-based victimization and SWB. Among students in lower academic levels, antisocial media exposure significantly predicted teacher victimization, while peer victimization was a strong and consistent predictor of diminished well-being. This pattern suggests that students in transitional academic stages, often marked by developmental sensitivity and

shifting social expectations, may be particularly vulnerable to both digital and relational stressors. In contrast, among students in higher academic levels, teacher victimization not only predicted peer victimization but also directly undermined SWB, indicating a more entrenched and institutionalized pattern of cascading victimization in the final year of secondary school. These findings extend the socioecological argument that distal digital influences are not merely external stressors but are internalized and enacted through proximal relational dynamics within school environments (Bronfenbrenner and Morris, 1998; Hong and Espelage, 2012). The school microsystem, in this context, becomes a conduit through which digital toxicity is translated into interpersonal harm, reinforcing the need to conceptualize media effects as embedded within broader social ecologies.

Nonetheless, this study has several limitations that warrant consideration. First, the use of self-reported measures may introduce social desirability bias, potentially affecting the accuracy of participants' responses. Second, the cross-sectional design limits the ability to establish causal relationships or determine the directionality of effects among variables. Third, the sample was drawn from a relatively small number of schools, which may restrict the generalizability of the findings to broader populations. Fourth, the study did not account for potentially influential factors, such as family environment, personality traits or community-level stressors, which may also shape students' well-being. Future research should consider longitudinal designs and more diverse samples to deepen the understanding of these dynamics and explore additional contextual variables. First, the use of self-reported measures may introduce social desirability bias, which may potentially affect the accuracy of participants' responses. However, a high level of confidentiality and privacy was accorded to participants during data collection, which possibly reduced the introduction of self-desirability bias. Future research should adopt longitudinal and mixed-method designs to capture the evolving nature of toxic media exposure and its relational pathways over time. Additionally, studies should explore how structural factors, such as school type, digital access and cultural norms, moderate these effects, particularly in under-researched Global South settings.

### *Research and theoretical implication*

This study offers several important contributions to the literature on digital media and adolescent well-being. By demonstrating that the effects of antisocial media exposure are mediated through teacher and peer victimization, it challenges dominant linear models that assume direct poor well-being from media exposure. Instead, it foregrounds relational mechanisms within the school environment as critical pathways through which antisocial media exposure is internalized and enacted. The identification of a significant serial mediation model, wherein antisocial media exposure leads to teacher victimization, which then facilitates peer victimization and ultimately erodes well-being, provides a robust empirical foundation for future longitudinal and intervention-based research. Theoretically, the study reinforces and extends Bronfenbrenner's ecological systems theory by illustrating how distal exosystemic forces, such as digital media, exert influence through interactions within the microsystem, particularly student–teacher and peer relationships (Bronfenbrenner and Morris, 1998). The findings affirm that antisocial media exposure infiltrates individual well-being through the social architecture of the school, where hierarchical (teacher–student) and lateral (peer–peer) victimization processes interact to shape developmental

outcomes. In the context of resource-constrained Nigerian schools, feasible interventions could include low-cost peer mentorship initiatives that foster prosocial behavior and short, skills-based teacher workshops on emotion regulation and nonpunitive classroom management. Such targeted, context-sensitive approaches can strengthen teacher–student relationships and reduce victimization without requiring extensive institutional resources.

Moreover, the intersectional analysis of gender and academic level reveals that these pathways are not uniform but are shaped by sociocultural norms and developmental timing. For instance, the heightened vulnerability of female students to peer victimization and the stronger link between media exposure and teacher victimization among males underscore the need for gender-responsive theoretical models. Similarly, the differentiated effects across academic levels suggest that developmental stage plays a critical role in how digital and relational stressors are experienced and internalized. Future research should endeavor to examine digital literacy, frequency of social media use, prior mental health issues and how they could influence the pathways proposed in our model. These insights call for a more dynamic and context-sensitive application of ecological theory in digital media research, particularly in Global South contexts where structural and cultural variables may amplify or buffer these effects.

To translate these findings into practice, schools should implement relational climate assessments to monitor student–teacher and peer dynamics, especially in contexts of digital stress. Teacher training programs must incorporate modules on recognizing and responding to media-related aggression, equipping educators to intervene early in relational victimization. Peer support systems and restorative practices should be embedded into school culture to reduce lateral aggression and promote inclusive norms.

### Conclusion

This study underscores the complex pathways through which antisocial media exposure influences the SWB of students. Rather than operating as a direct negative stressor, antisocial media exposure exerts its effects through relational and institutional dynamics, specifically through teacher and peer victimization within school environments. The findings illuminate how gender and academic level shape these pathways, revealing distinct vulnerabilities and reinforcing the need for intersectional and developmentally sensitive interventions. By situating antisocial media exposure within the socioecological framework and emphasizing the role of school-based relationships, this study contributes to a deeper understanding of how media toxicity is socially mediated and developmentally consequential. It calls for more nuanced, context-aware approaches in both research and practice, particularly in under-resourced educational settings, where the well-being costs of digital engagement are often compounded by institutional neglect and relational problems.

**Open peer review.** To view the open peer review materials for this article, please visit http://doi.org/10.1017/gmh.2025.10116.

**Author contribution.** TYA: Conceptualization, data curation, funding acquisition, investigation, visualization, formal analysis, project administration, writing – review and editing, writing – original draft. SOI: Project administration, methodology, supervision, writing – review and editing. DS: Supervision, methodology, visualization, writing – review and editing. OOA: Methodology, visualization, writing – review and editing. TKO: Project administration, methodology, visualization, supervision, writing – review and editing. ASN: Methodology,

visualization, writing – review and editing. ADO: Project administration, methodology, visualization, supervision, writing – review and editing.

**Competing interests.** The authors declare none.

**Ethical consideration.** This study received ethical approval from the Oyo State Ministry of Health (Ref no: AD13/479/715A) and the Oyo State Health Research Ethics Committee (NHREC/OYOSHRIEC/10/11/12), as part of a broader investigation into adverse life experiences mediated by environmental stressors. Informed consent was obtained from parents or legal guardians of all participating students before data collection. Additionally, institutional approval was secured from the heads of each participating school, ensuring compliance with ethical standards for research involving minors and school-based populations.

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
