## [Reviewer Report]

Thank you for the opportunity to review this manuscript! This study investigates a serial mediation model linking exposure to antisocial media exposure and subjective well-being among young people in Nigeria. This study may provide a perspective to the literature of youth subjective well-being in the digital era.

However, this manuscript at the current stage still has limitations. My suggestions to improve this study are provided below for the author(s) information.

1. Rationales behind serial mediation pathways: The authors should strengthen their justifications about the serial mediation model. Readers may easily challenge that the mechanisms among the focal variables may be parallel mediation pathways as well, given that studies have shown that peer victimization could also lead to teacher victimization. In particular, this study used a cross-sectional design to investigate a serial mediation model, which needs more powerful elaborations about the relationships between mediators.

2. Theoretical framework: this study used ecological theoretical framework to develop their hypotheses. This framework also argues the important roles of family factors in shaping personal mental health and well-being. However, family factors are missed in empirical models or discussions throughout the manuscript. I suggest that the authors put some efforts to analyse or discuss this perspective.

3. Group differences: this study explored group differences in gender and academic levels regarding the proposed mediation model. However, the literature review did not provide solid justifications about the rationales of considering these group differences. I suggest that the authors use a separate section to justify reasons about conducting group comparisons.

4. Measures: antisocial media exposure was measured by an eight-item scale in this study, while it was treated as an observed variable unlike other latent variables in the empirical model. The authors need to justify considerations behind this approach, otherwise, antisocial media exposure should be included as a latent variable in the model.

5. Measures: the measurement of academic levels was missing. The authors need to justify how they decided the cut-off value of academic levels, otherwise, the current approach seems arbitrary.

6. Methods: all variables were extracted from single-informant and self-report responses, which may induce common methods bias. The authors need to report how they addressed this issue.

7. Figure 1: Living arrangement was not included as a covariate in the empirical model, while Table 1 and Table 2 included information about this variable. The authors need to justify why they excluded this control variable.

8. Sample: emerging adults tend to indicate people aged above 18 years old. The current sample is a mixture of adolescents and emerging adults. Adolescents and emerging adults have different developmental stages and performances. The authors should consider focusing on a specific group, either adolescents or emerging adults.

9. Implication: the authors overlooked practical implications from their empirical findings. Indeed, this may be important to communicate their scientific progress with the public. Highlighting practical implications may distinguish contributions of this study from previous studies, given that theoretical contributions of the current manuscript are not strongly elaborated.

10. Limitations: the discussions about limitations and future research directions are weak. The authors may need to provide detailed discussions with literature support, which may strengthen scientific values of this study.

---

## [Reviewer Report]

This manuscript addresses a timely and understudied topic: the indirect pathways linking antisocial media exposure to subjective well-being (SWB) among emerging adults, with a focus on school-based victimization (peer and teacher) as mediators. The study’s strength lies in its application of Bronfenbrenner’s socioecological framework to contextualize digital harm within real-world school dynamics, its attention to intersectional differences (gender, academic level), and its empirical rigor like using PLS-SEM, bootstrapping for mediation. By focusing on a Nigerian sample—an underrepresented context in global digital mental health research—the work fills a critical gap in understanding how structural and cultural factors (e.g., resource-constrained schools) shape media-victimization-well-being linkages.

That said, several areas require refinement to strengthen the manuscript’s clarity, generalizability, and theoretical contribution. Below are detailed comments organized by key sections, followed by actionable suggestions.

1. The introduction notes the study focuses on Nigerian emerging adults but provides little context for why this population is uniquely vulnerable. For example, are there country-specific trends in digital media access (e.g., high social media use among adolescents but limited digital literacy programs)? Or structural challenges in Nigerian schools (e.g., overcrowded classrooms, limited teacher training) that might amplify victimization? Without this, readers cannot fully grasp the study’s contextual significance.

2. The introduction claims “prior research has yet to fully explore how exposure to such contents contributes to dual victimization from both peers and teachers,” but it does not cite specific studies that have examined single forms of victimization (e.g., peer victimization alone) to highlight the gap in dual-victimization research.

3. The scale is adapted from the California School Climate and Safety Survey (CSCSS), but the authors do not report whether they validated the adapted version for a Nigerian sample. Cross-cultural adaptation of scales requires testing for cultural relevance (e.g., are items like “Teachers touched you unnecessarily” consistent with Nigerian school norms?) and psychometric properties (e.g., Cronbach’s α for the adapted scale—Table 3 lists α=0.850 for teacher victimization but does not clarify if this is for the original or adapted scale).

4. The authors control for age, parental education, and living arrangement but do not explain why other potential confounders (e.g., digital literacy, frequency of social media use, prior mental health issues) were excluded. For example, a student with low digital literacy might be more likely to encounter antisocial content and struggle with peer relationships, which could confound the mediation pathway.

5. The authors note that SWB has a low R² (0.085, meaning 8.5% of variance is explained by the model) but do not discuss what this implies. For example, are unmeasured factors (e.g., family support, community violence) more influential for Nigerian students’ SWB? This is a critical limitation that deserves attention in the results or discussion.

6. The discussion calls for “targeted interventions that address both teacher and peer victimization” but provides few concrete examples tailored to Nigerian schools. For instance, resource-constrained schools may not have the budget for large-scale programs—what low-cost interventions (e.g., peer mentorship, brief teacher training workshops) could be feasible?

---

## [Editor Report]

Dear authors, 

Two experts have carefully reviewed your manuscript. When they agreed this is a timely topic with great interest, it still requires a major revision, including justification of the serial mediation model and the theoretical framework, data analyses, result interpretation. I think both the reviewers have provided very constructive feedback that should carefully considered in order to improve the quality of this manuscript. I am looking forward to seeing the revised manuscript.

---

## [Reviewer Report]

The authors have worked hard to address all the reviewers' comments. I have no further comment for their revision.